# Novel Immunotherapeutic Approaches to Treating HPV-Related Head and Neck Cancer

**DOI:** 10.3390/cancers15071959

**Published:** 2023-03-24

**Authors:** Nabil F. Saba, Saagar Pamulapati, Bhamini Patel, Mayur Mody, Primož Strojan, Robert Takes, Antti A. Mäkitie, Oded Cohen, Pia Pace-Asciak, Jan B. Vermorken, Carol Bradford, Arlene Forastiere, Yong Teng, Andreas Wieland, Alfio Ferlito

**Affiliations:** 1Department of Hematology and Medical Oncology, Winship Cancer Institute, Emory University, Atlanta, GA 30322, USA; 2Internal Medicine Program, Mercyhealth, Rockford, IL 61114, USA; 3Department of Internal Medicine, Emory University, Atlanta, GA 30307, USA; 4Hematology and Oncology Program, AdventHealth Medical Group, Calhoun, GA 30701, USA; 5Department of Radiation Oncology, Institute of Oncology, 1000 Ljubljana, Slovenia; 6Department of Otolaryngology and Head and Neck Surgery, Radboud University Medical Center, 6525 Nijmegen, The Netherlands; 7Department of Otorhinolaryngology—Head and Neck Surgery, Helsinki University Hospital, University of Helsinki, Research Program in Systems Oncology, FI-00014 Helsinki, Finland; 8Department of Otolaryngology, Ben Gurion University of the Negev, Soroka Medical Center, Be’er Sheva 84-101, Israel; 9Department of Otolarynology—Head and Neck Surgery, University of Toronto, Toronto, ON M5S 1A1, Canada; 10Department of Medical Oncology, Antwerp University Hospital, 2650 Edegem, Belgium; 11Faculty of Medicine and Health Sciences, University of Antwerp, 2610 Antwerp, Belgium; 12Department of Otolaryngology, The Ohio State University, Columbus, OH 43212, USA; 13Department of Oncology, The Johns Hopkins University School of Medicine, Baltimore, MD 21287, USA; 14Department of Microbial Infection and Immunity, The Ohio State University, Columbus, OH 43210, USA; 15Pelotonia Institute for Immuno-Oncology, The Ohio State University Comprehensive Cancer Center, The Ohio State University, Columbus, OH 43210, USA; 16Coordinator of the International Head and Neck Scientific Group, 35100 Padua, Italy

**Keywords:** human papillomavirus, HPV, immunotherapy, vaccine therapy, immuno-oncology

## Abstract

**Simple Summary:**

Recent advances in immunotherapeutic approaches have spanned different tumor types including head and neck cancer. HPV-related head and Neck Cancers are virally driven tumors with notable immune characteristics and recent advances in immune targeting that desrve a closer examination. We aimed in this comprehensive review to offer a description of recent advances in immunotherapeutic applications in this disease.

**Abstract:**

Head and neck cancer (HNC) is the seventh most common malignancy, with oropharyngeal squamous cell carcinoma (OPSCC) accounting for a majority of cases in the western world. While HNC accounts for only 5% of all cancers in the United States, the incidence of a subset of OPSCC caused by human papillomavirus (HPV) is increasing rapidly. The treatment for OPSCC is multifaceted, with a recently emerging focus on immunotherapeutic approaches. With the increased incidence of HPV-related OPSCC and the approval of immunotherapy in the management of recurrent and metastatic HNC, there has been rising interest in exploring the role of immunotherapy in the treatment of HPV-related OPSCC specifically. The immune microenvironment in HPV-related disease is distinct from that in HPV-negative OPSCC, which has prompted further research into various immunotherapeutics. This review focuses on HPV-related OPSCC, its immune characteristics, and current challenges and future opportunities for immunotherapeutic applications in this virus-driven cancer.

## 1. Introduction

Head and neck cancer (HNC) remains a highly prevalent disease and is now ranked the seventh most common malignancy globally. In the United States, there are almost 66,000 new cases yearly [1]. Oropharyngeal squamous cell carcinoma (OPSCC) accounts for over half of HNC. Historically, tobacco and alcohol use were the primary risk factors in the development of HNC, including OPSCC. However, despite a significant decrease in tobacco use over the last several decades, the incidence rates of OPSCC have continued to rise. This has been largely attributed to the increased prevalence of human papillomavirus (HPV) infection. Data from the Surveillance Epidemiology and End Results (SEER) has demonstrated an increased incidence of HPV-related OPSCC from 16.3% in the 1980s to 72.7% in the early 2000s. This trend is also noted in HNC incidence overall [2]. Additional SEER data have also demonstrated worse cancer-specific mortality in non-white and uninsured patients with HPV-positive OPSCC. This difference was not observed in non-OPSCC or HPV-negative diseases, suggesting that socioeconomic disparities contribute to worse outcomes despite a clinically favorable disease [3]. Increased incidence has been noted globally as well, including countries in Europe, Southeast Asia, and South America, with a pooled burden of HPV-positive OPSCC of about 33%. Additionally, this increase in prevalence has occurred gradually, alluding to the increased burden of HPV-positive OPSCC worldwide [4]. It is crucial to understand this change in etiology for OPSCC as it can significantly impact prognosis and treatment strategies for patients.

HPV is well-known to be a predominant factor in the development of cervical cancer and has also been implicated in anal, penile, vaginal, and oropharyngeal carcinomas, as it can affect the squamous epithelium in these sites. HPV is relatively common within the general population, and most individuals are able to clear the virus or remain largely asymptomatic [5]. However, high-risk strains such as HPV type 16, 18, or 31 can integrate themselves within human nuclear DNA, resulting in a severely dysregulated cell cycle and transformation. The viral oncogenes found in these high-risk strains, E6 and E7, encode regulatory proteins that play a crucial role in carcinogenesis. E6 binds protein p53, a tumor suppressor protein, thereby promoting its degradation, and E7 binds the retinoblastoma (Rb) protein leading to its inactivation. The loss of both protein p53 and Rb proteins leads to unregulated cell division, genomic instability, and loss of apoptosis [6,7].

Although primarily thought to be associated with middle-aged, white men, more recent data suggest increasing rates of HPV-positive OPSCC among elderly (65 and above) patients in the United States [8,9,10]. HPV-positive status is associated with notable differences in clinical presentation, such as more extensive nodal disease and smaller primary tumors [11]. A crucial difference between HPV-related and unrelated OPSCC is its significantly improved overall survival (OS) and higher cure rates. A 40% reduction in death from all causes and a 60% reduction in death due to cancer have been reported in HPV-related versus unrelated diseases [12]. A large, randomized, multi-institutional study of 721 patients with OPSCC found improved OS and progression-free survival (PFS) in HPV-positive patients (*p* < 0.001) [13]. The study demonstrated a 3-year OS of 82.4% vs. 57.1% and PFS of 73.7% and 43.4% in HPV-related vs. unrelated diseases, respectively. This survival advantage persisted even after adjustment for age, race, performance status, tumor/nodal stage, and the number of years of smoking. A recent update with a median follow-up of 7.9 years continues to demonstrate improved OS in HPV-related disease (70.9% vs. 30.2%) [14]. Several other studies have shown similar results with varying regimens of radiotherapy (RT), chemotherapeutic agents, and targeted therapy [13,14,15,16,17].

Studies with a higher prevalence of HPV-positive OPSCC patients are showing even higher rates of disease-specific and overall survival. In a phase II clinical trial of intensity-modulated chemoradiotherapy to reduce dysphagia in advanced-stage OPSCC, 90% of patients were HPV-positive, and 3-year disease-specific survival was 88% [18]. Transoral surgical approaches are also showing favorable outcomes in HPV-positive OPSCC. Excellent 3-year disease-specific survival rates of 88% were observed in a prospectively assembled database of 204 advanced OPSCC patients (90% were HPV-positive) who underwent transoral laser surgery for OPSCC with risk-based adjuvant radiotherapy or chemoradiotherapy [19]. A randomized phase II trial of transoral surgery followed by low-dose or standard-dose intensity-modulated radiotherapy was conducted in patients with resectable p16-positive locally advanced oropharynx cancer. Two-year survival rates ranging from 90.5% to 95.9% were observed in low, intermediate, and high-risk patients, suggesting that the regimen of transoral surgery and low-dose radiotherapy is considered worthy of further study [17]. Ongoing research regarding prognosis and HPV status has led to a fundamental change in the American Joint Committee on Cancer (AJCC) staging system for OPSCC, with HPV status incorporated for better risk stratification [20].

The reasons behind the survival benefit demonstrated in HPV-related diseases are not entirely clear at present. One theory suggests the improved locoregional control may be due to increased radiosensitivity [21]. Additionally, HPV-related OPSCC has decreased recurrence rates, possibly secondary to the lack of clear presence of field cancerization as explained by the lack of transcriptionally active HPV in surrounding mucosa [22]. Lastly, HPV-related OPSCC tends to have decreased rates of second primaries, which may be due to the lack of traditional risk factors such as tobacco and alcohol use [23]. Despite the improved survival rates, almost 10–25% of patients with HPV-related OPSCC develop disease recurrence often within the first two to five years after diagnosis. Factors such as stage at presentation (AJCC 8 T4 or N3 clinical stage), tobacco use, and tumor-related hypoxia are possible identifiers for worse clinical outcomes [24].

Despite phenotypic similarities between HPV-related and unrelated OPSCC, it is vital to understand their underlying differences as this can shed light on their respective biology and potentially impact treatment strategies. This review will outline the management principles for HPV-related disease, discuss immunological differences when compared to HPV-unrelated OPSCC, and consider current and future targets for immunotherapy. Additionally, integrating further primary prevention strategies with these novel therapeutic strategies will enable us to move toward the eradication of HPV-related OPSCC. The aims of this review include detailing HPV-related OPSCC, its immune characteristics, and current challenges and future opportunities for immunotherapeutic applications in this virus-driven cancer.

## 2. The Influence of HPV Status on the HNC Immune Microenvironment

In order to understand the particulars of immune evasion by HPV-related OPSCC, the general immune microenvironment of HNC must first be delineated. The interplay between the tumor and the immune system is complex; various mechanisms allow for cancer immunity [25]. Destruction of cancer cells is typically facilitated by cytotoxic T cells that recognize tumor-specific antigens. Professional antigen-presenting cells (APC) presenting tumor-specific antigens initially prime naive T cells resulting in their activation, proliferation, and acquisition of cytotoxic capabilities. As tumor cells continue to mutate throughout their life, they will generate additional tumor antigens that, upon tumor cell lysis, can be presented by APCs and prime additional T cells, thus broadening the antitumor immune response [26]. Of note, HNC is particularly impacted by mutagenesis and mutational burden, a point of consideration in regard to the clinical benefit of immunotherapeutic approaches [27].

HPV-related OPSCC has demonstrated a distinct immune microenvironment. One difference is that, although the expressed viral oncoproteins are foreign and thus exquisite immunological targets, the viral oncoproteins E6 and E7 can inhibit the presentation of tumor antigens on tumor cells through the downregulation of major histocompatibility complex (MHC) class I and class II molecules [28,29]. This interference in antigen presentation is one way in which the HPV-related OPSCC immune microenvironment is distinct. Other aspects of HPV-related OPSCC immune evasion, also observed in other tumors, include aberrant regulation of immunosuppressive cytokines and signal transducer and activator of transcription (STAT) proteins, as well as the dysregulation of interferon-gamma (IFNγ)-producing immune effector cells [30,31]. The dysregulation of these pathways is key to how tumor cells can persist even in the face of substantial antitumor immune responses.

HPV-related OPSCC also exhibits increased activation of immune infiltrates compared to HPV-unrelated disease. Multiple studies have indicated increased activation of T cells in HPV-related disease as evidenced by increased levels of perforin and granzymes, which are integral to the induction of apoptosis through cytotoxic T cells [32,33]. Other studies have also demonstrated increased infiltration of regulatory T (Treg) cells and natural killer (NK) cells in HPV-related disease [34,35]. In addition to T cell activation, studies on HPV-related tissue samples also demonstrated the presence of antibody-secreting cells producing HPV-specific antibodies directly within the tumor microenvironment (TME) [36]. Importantly, the antibodies produced in the TME targeted not only the viral oncoproteins E6 and E7 but also, to a major extent, E2, which is expressed in a majority of HPV-related OPSCC cases due to the episomal maintenance of the HPV genome [37]. A depiction of the TME associated with HPV-positive disease is illustrated in Figure 1. These findings demonstrate the presence of an ongoing humoral immune response in HPV-related HNC and highlight the need to further study B cells as a potential therapeutic target [36].

The increased tumor infiltration by a variety of lymphocytes is particularly important as this has been correlated with improved prognosis [38,39]. When patients with HPV-related OPSCC are stratified into groups with high and low levels of tumor-infiltrating lymphocytes (TILs), patients in the high TIL group had a significantly improved three-year survival rate compared to the low TIL group [40]. These findings were reinforced by subsequent studies showing that TIL levels could be used as risk stratification for high and low-risk HPV-related OPSCC [41,42]. Additionally, when comparing HPV-unrelated to HPV-related OPSCC with low TIL levels, there was no significant change in survival [41]. These findings suggest that increased TIL infiltration may partially explain the difference in prognosis between HPV-related and unrelated diseases [43,44]. Of note, a recent study showed that the TME not only contains substantial numbers of HPV-specific CD8 T cells but, more importantly, also a distinct subset of stem-like cells among HPV-reactive T cells that possess proliferative potential [45]. These stem-like CD8 T cells, characterized by co-expression of the inhibitory receptor PD-1 and the transcription factor TCF-1, have been shown to provide the proliferative burst upon PD-1 pathway blockade in preclinical models, supporting the use of immunotherapeutic agents in HPV-related OPSCC [46,47,48]. Furthermore, the analysis of intratumoral CD8 T cells regarding their fine antigen specificity also revealed substantial responses against E2 and E5, suggesting that these proteins should be considered potential targets for vaccine therapeutics in addition to E6 and E7 antigens [45].

Another explanation for the difference in prognosis has been attributed to the increased expression of PD-L1 in HPV-positive OPSCC [49,50]. Approximately half of OPSCC express PD-L1, with higher expression, noted in HPV-positive disease [51]. The PD-1/PD-L1 axis and its regulation are complex, and tumors widely employ this immuno-inhibitory axis to evade destruction. The engagement of PD-1 on T cells by PD-L1 expressed on tumor cells or APCs results in the dampening of T cell receptor signaling, which allows for the moderation of immune responses in order to prevent overactivity [52]. The difference in PD-L1 expression and the other unique aspects of the HPV-related OPSCC immune microenvironment outlined above are critical to understanding how HPV-related OPSCC may respond differently to novel immunotherapeutic approaches.

## 3. Immunotherapy

Treatment modalities for locally advanced (LA) and recurrent/metastatic (RM) HNC have drastically changed over the past several decades. Traditionally, the treatment of LA HNC, regardless of HPV status, entails a combined modality approach involving surgical resection, radiotherapy, and/or chemoradiotherapy (CRT) [54]. The current standard regimen includes high-dose cisplatin (100 mg/m^2^) every three weeks with concurrent radiation. A large meta-analysis demonstrated the benefit of CRT compared to induction chemotherapy and cemented single-agent, platinum-based therapy as the preferred chemotherapeutic option [55]. However, given the overall improved outcomes in HPV-positive patients in a generally younger and otherwise healthy patient population, there is a need for less toxic regimens to better preserve the quality of life. Significant research has been conducted to find alternative dosing strategies, less toxic chemotherapeutic agents, and alterations in radiation delivery. A possible alternative includes using weekly, low-dose cisplatin, which has shown improved adherence and lower rates of grade III and IV adverse events while maintaining similar survival benefits [56]. A retrospective review found similar findings in patients with HPV-related disease and reported no difference in 3-year OS or recurrence-free survival with three-weekly compared to weekly cisplatin [57]. Additionally, some data suggest similar overall outcomes in patients with LA HNC with carboplatin or taxane-based chemotherapy agents, which could be viable alternatives to patients unable to tolerate cisplatin [58,59].

In addition to cytotoxics, the inhibition of epidermal growth factor receptor (EGFR) through targeted monoclonal antibodies such as cetuximab significantly altered the treatment landscape in the definitive management of LA and RM HNC. Two landmark trials led to the approval of cetuximab in conjunction with radiation therapy as first-line treatment in LA in addition to platinum-based chemotherapy in RM HNC [60,61]. Several reports since then have demonstrated worse outcomes with single-agent anti-EGFR therapy in HPV-positive tumors compared to HPV-negative [62]. This data indicates that cetuximab-based therapy and EGFR targeting may not be an optimal strategy in HPV-related OPSCC.

### 3.1. Immune Checkpoint Inhibitors

Immune checkpoint inhibitors (ICIs) have shown unprecedented promise in the treatment of RM HNC, including HPV-related OPSCC, primarily through their impact on immune escape. One mechanism for tumor immune evasion is through the PD-1/PD-L1 axis [63]. Tumor cells upregulate PD-L1 expression, which leads to increased engagement of the PD-1 receptor on T cells, thus inhibiting immune-mediated tumor cell killing. ICIs block this interaction by binding to either the PD-1 receptor or PD-L1 receptor and thereby reinvigorate tumor-specific T cell responses [64].

The anti-PD-1 antibody pembrolizumab initially demonstrated success in KEYNOTE-012, a phase Ib trial assessing safety and efficacy in an expansion cohort with a reported overall response rate (ORR) of 18%, 6-month progression-free survival (PFS) and OS of 23% and 59%, respectively. However, when assessing HPV-related and unrelated diseases, the ORR was 32% vs. 14%, 6-month PFS was 37% vs. 20%, and 6-month OS was 70% vs. 56%, respectively [65]. This suggests an improved response in HPV-related disease, but this association should be taken with caution given the small cohort of patients and heterogeneity in determining HPV status. A phase II study, KEYNOTE-055, with a similar size and patient population, demonstrated comparable results within the overall population but no reported difference in ORR, PFS, or median OS based on HPV status [66]. The KEYNOTE-040 was a multicenter randomized, phase III trial that compared pembrolizumab to methotrexate, docetaxel, or cetuximab monotherapy in 495 patients. In the intention-to-treat analysis, the median OS was 8.4 months in pembrolizumab vs. 6.9 months in the standard of care (SOC) arm, with decreased adverse events in the pembrolizumab arm. Similar to KEYNOTE-055, there was no observable difference in OS based on HPV-status between the two groups [67]. KEYNOTE-048, which demonstrated statistically significant improvement in median OS and led to the approval of pembrolizumab as a first-line agent for RM HNC disease, randomized 882 patients to pembrolizumab monotherapy, pembrolizumab with platinum-based chemotherapy, or cetuximab with chemotherapy (EXTREME). Several subgroup analyses were conducted based on PD-L1 combined positive score (CPS). Pembrolizumab monotherapy significantly improved OS compared to the EXTREME regimen in patients with a CPS of 20 or more (14.9 months vs. 10.7 months) but was non-inferior in the total population. However, pembrolizumab with chemotherapy compared to the EXTREME regimen significantly improved OS in the total population (13.0 vs. 10.7 months), CPS of 20 or more (14.7 vs. 11.0 months), and CPS of 1 or more (13.6 vs. 10.4 months) [51].

Nivolumab, a human IgG4 anti-PD-1 monoclonal antibody, demonstrated success in the treatment of RM HNC in the CheckMate-141 study. CheckMate-141 was the first phase III randomized trial showing superior survival of an immunotherapeutic agent over cytotoxic chemotherapy in patients with RM HNC, including HPV-related OPSCC. CheckMate-141 assessed 361 patients who failed platinum-based therapy in the recurrent setting or progressed within six months of platinum-based definitive chemotherapy. The patients were randomized to nivolumab every two weeks or SOC and received either methotrexate, docetaxel, or cetuximab. The study met its primary endpoint with significantly improved median OS in the nivolumab arm (7.5 vs. 5.1 months). Patients receiving nivolumab also maintained their quality-of-life scores, whereas the SOC arm had worse outcomes with treatment [68]. The improved median OS rate persisted at a 2-year follow-up in the nivolumab vs. SOC arms (16.9% vs. 6.6%). This difference was maintained regardless of HPV-status and age and persisted for patients who received first-line therapy with nivolumab [69]. The CheckMate-651 study randomized a total of 947 patients to the combination of nivolumab and ipilimumab versus the EXTREME regimen. Ipilimumab is a monoclonal antibody against cytotoxic T lymphocyte antigen-4 (CTLA-4), which is another mechanism for immune escape by tumor cells. The study randomized a total of 947 patients and, at a minimum follow-up of 27.3 months, did not find a statistically significant improvement in OS regardless of CPS score or HPV status when compared to the EXTREME regimen. However, grade 3 and 4 adverse events occurred in 28.2% of patients in the combination arm and 70.7% in the EXTREME arm [70]. Although CheckMate-651 did not meet its primary endpoint of OS, the combination arm demonstrated a more favorable safety profile.

Durvalumab is another well-studied PD-L1 monoclonal antibody in the treatment of RM HNC. The HAWK trial was a multi-institutional, single-arm, phase II study assessing durvalumab in platinum-refractory RM disease. The study included 112 immunotherapy naive patients with high PD-L1 expression and demonstrated an ORR of 16.2%, PFS of 2.1 months, and median OS of 7.1 months. A secondary analysis performed on all HPV-positive tumor sites demonstrated an ORR of 29.4% (10/34) in HPV-positive versus 10.8% (10/65) in HPV-negative tumors. Additionally, HPV-positive tumors had a numerically higher PFS and median OS compared to HPV-negative tumors. However, it should be noted that this population included solely patients with high PD-L1 expression, which may be a contributing factor to improved survival rates [71]. The CONDOR trial was a randomized, phase II study that assessed durvalumab monotherapy, tremelimumab (a CTLA-4 monoclonal antibody) monotherapy, and combination tremelimumab and durvalumab in 267 patients with refractory RM HNC and low/negative PD-L1 tumor expression. The study did not demonstrate a difference in OS or side effect profile between the three trial arms. Among HPV-positive tumors, the ORR was 5.4% in the combination arm and 16.7% in the durvalumab arm, but both results had wide confidence intervals [63]. In the EAGLE study, 736 patients were randomized to treatment with durvalumab, durvalumab plus tremelimumab, or SOC chemotherapy (taxane, methotrexate, cetuximab, or fluoropyrimidine regimen). There was no statistically significant benefit in OS or PFS between the durvalumab arms and SOC. However, numerically there were higher response rates at 12 and 24 months in the durvalumab arms [72].

It is clear, based on the aforementioned, that ICIs provide an improved OS with minimal toxicity in HPV-related and unrelated HNC, allowing patients to preserve both quantity and quality of life. Given the improved survival in patients with OPSCC compared to other head and neck tumor sites, there was reason to believe immunotherapy would enhance this survival benefit. However, as discussed above, there are conflicting data regarding if immunotherapy favorably impacts HPV-positive tumors. A meta-analysis that assessed nine studies with PD-1/PD-L1 inhibition provided further answers. The study performed a subgroup analysis for pooled ORR and demonstrated an ORR of 18.8% in HPV-positive patients versus 12.2% in HPV-negative patients. The study also suggested a 56% higher chance of achieving ORR in HPV-positive patients than in HPV-negative patients. However, these results did not meet statistical significance [73]. Further research needs to be conducted to determine the impact of PD-1/PD-L1 targeted immunotherapy on HPV-positive tumors.

Of note is the potential of other immunomodulatory approaches in the realm of immunostimulatory options and novel combinatorial approaches in treating recurrent and metastatic head and neck squamous cell carcinomas. These may, in the future, alter the therapeutic landscape of HPV-related OPSCC. Examples include targeting IDO1 [74], TLR8 agonists [75], OX40 [76], and B7-H3 [77], the detailing of which are beyond the scope of this review. Major trials regarding ICIs and HPV-related OPSCC are demonstrated in Table 1 [51,63,65,66,67,69,70,71,72]. 

### 3.2. T-Cell Associated Therapeutics

T cells are an integral part of the immune system, working to detect foreign agents and facilitate adaptive immune responses. Thus, they play a crucial role in mounting the body’s antitumor response. There are several ongoing studies focusing on formulating T cells that target specific antigens to provide therapeutic benefits.

A specific form of T cell engineering known as adoptive cell-based therapy is currently being investigated. This therapy isolates target T cells from the patient, transduces them in vivo to specifically target various antigens, and then transplants them back into the host after being genetically modified. Adoptive cell-based therapy has been previously administered in patients with B cell malignancies and metastatic melanoma [78,79]. A phase II clinical trial (NCT01585428) evaluated autologous T cell transfusion in 29 patients with HPV-associated cancers. The study found two partial responses and one stable disease when assessing the eleven noncervical HPV-related tumors. Of note, one of those patients who exhibited a partial response had oropharyngeal cancer and attained a partial response of 5 months duration [80].

MHC molecules serve as an intermediary between tumor antigens and T cell receptors on antigen-specific T cells. Some particular tumor antigens for HPV include E6 and E7, which are oncoproteins inside HPV-positive cancer cells and whose peptides can be presented on MHCs. Researchers are now focusing on these proteins as optimal targets, as they are absent from healthy human tissues. Therapy targeting E7 oncoprotein has been validated in a preclinical study in mice, which revealed that genetically modified T cells inhibited tumor growth, prolonged survival, and promoted immunological memory in HPV-related cervical and oropharyngeal cancer [81]. A phase I/II study (NCT02858310) investigated targeting the E7 oncoprotein. The preliminary data reports a total of twelve heavily pre-treated patients with HPV-related cancers, of which four participants were diagnosed with HNC. All four patients with HNC had PD1 inhibitor-refractory disease. Of the four patients with HPV-positive HNC, two demonstrated a partial response, and two had stable disease [82]. Another phase I study (NCT03578406) evaluates T cells genetically modified with T cell receptors targeting E6 for patients with HPV-related cervical or HNC. Another question is whether there is a benefit to administering modified T cells in conjunction with a PD-1 antagonist. The hypothesis is that suppressing the PD-L1/PD1 axis will further increase the efficacy of modified T lymphocytes in HPV-related malignancies [83]. An additional phase I/phase II trial (NCT02280811) evaluated T-cell receptor gene therapy directed against E6 in HPV-related malignancies. Of note, of the 12 patients included in the study, only one was diagnosed with HNC. The study reported two patients with partial responses and four patients with stable disease, which included the patient with HNC. These results indicate that autologous T cell therapy focusing on E6 engendered anti-HPV tumor effects [84].

Although initial efforts are promising, multiple impediments have been established that need to be further elucidated. One concern is the risk of T cell receptor mispairing, which involves the formation of a hybrid T cell receptor by pairing the endogenous with the genetically engineered T cell receptor chains. This mispairing has been shown to cause graft vs. host disease in previous animal studies [85]. Another risk of such therapy is nonspecific cytotoxicity, caused by an incorrect attack of the modified T cells against nonmalignant cells which express similar antigens to those being targeted. Previous studies using engineered T cells for melanoma and colorectal cancer showed marked adverse events such as fatal cardiotoxicity and severe colitis, respectively [86,87]. The most frequent risk of T cell immunotherapy is a cytokine storm, a potentially life-threatening condition characterized by high fevers, inflammation, myalgias, hypotension, dyspnea, or severe nausea [88]. Future studies to evaluate the risk of such adverse events are needed.

### 3.3. Immuno-STATs

Immuno-STATs (Selective Targeting and Alteration of T cells) are the most recent development utilizing T cells for cancer-targeted therapy. The Immuno-STAT is a modular fusion protein consisting of a tumor-associated human leukocyte antigen (HLA) peptide complex that delivers interleukin-2 to induce the proliferation of tumor-antigen-specific T cells. Selective targeting and delivery of the tumor antigen-specific T cells reduce systemic activation of the immune system, thereby avoiding toxicity [89].

A recent phase I trial (NCT03978689) studied CUE-101, an Immuno-STAT composed of an HLA presenting an HPV-16 E7 peptide and four molecules of IL-2, which binds to HPV-specific CD8 T cells and induces their activation and proliferation. The study assessed escalating doses of CUE-101 monotherapy or in combination with pembrolizumab based on previous PD-L1 treatment in patients with HPV-16+ recurrent or metastatic HNC. Therapy was provided every 3 weeks until disease progression or toxicity. CUE-101 demonstrated safety and tolerability with sustained increases in NK cells and Treg cells. Of the 14 evaluable patients in the monotherapy cohort, there was 1 partial response (7%) and 6 stable disease (43%). Of the 7 evaluable patients in the combination arm, there were 2 partial responses (29%) and 2 stable disease (29%) for >12 weeks. Given its target of E7, this modular fusion protein can be utilized across several different cancer types, including HPV-positive HNC. The FDA has now provided provisional approval for the use of CUE-101 at this time [90]. Novel trials concerning therapeutics targeting E6 and E7 oncoprotiens are summarized in Table 2 [80,82,83,84,90].

### 3.4. B-Cell Associated Therapeutics

Despite significant strides in understanding T cell action within the TME, there is still much to learn regarding antitumor immunity provided by B cells. The humoral immune response has several functions, including antigen presentation, cytokine release, and evoking antibody-dependent cellular cytotoxicity through antibody secretion. More importantly, B cells have been shown to exhibit several surface receptors, such as PD-1, PD-L1, and CTLA-4, which will be affected by current immunotherapeutic agents and might thus also contribute to their clinical activity [91].

The presence of increased B cell infiltration within HPV-related tumors has also been associated with a more favorable prognosis [92,93]. Studies assessing HPV-related TME have demonstrated the presence of antibody-secreting cells and B cells in well-organized structures within the tumor stroma. Additionally, HPV-specific antibodies were found to be actively secreted in situ and directed against the E2, E6, and E7 proteins [36]. Additionally, increased B cell germinal center formation has been observed after PD-1 blockade and radiation therapy, further suggesting the need to develop therapeutics targeting B cells [94].

## 4. Therapeutic Vaccination for Personalized Cancer Treatment

Another way to induce T cell responses in recognizing and destroying malignant cells is through personalized cancer vaccination therapy [95]. There are various vaccination platforms, including DNA, RNA, and peptide vaccines [96]. Cancer vaccination aims to promote T cell responses to specific tumor-associated antigens and tumor-specific antigen targets. The vaccines are designed in such a way that they elicit cell-mediated immunity rather than neutralizing antibodies, which is the case for most prophylactic vaccines [97]. As described earlier, HPV-associated HNC expresses the virus-derived E6 and E7 proteins that can be targeted by such therapy [12]. Interestingly, vaccine constructs that target both E6 and E7, rather than one single protein alone, have been shown to induce a more robust immune response [98].

Multiple phase I studies have demonstrated success in this endeavor. A phase I study (NCT00257738) evaluated the immunologic response of a peptide immunomodulator GL-0810 and MAGE-A3. A limited number of patients completed the dose-escalated vaccination course, of which 80% (4/5 patients) in the GL-8010 cohort and 67% (4/6 patients) in the MAGE-A3 cohort had an objective T cell and antibody response measured through IFN-γ ELISPOT and ELISA, respectively. Apart from mild reactions at injection sites such as erythema, pain, or pruritis, the treatment was overall well tolerated. Despite the clinically documented progression in all patients, these results suggested that the peptide vaccine was safe and could induce an immunologic response [99].

Another phase I study investigated the DNA vaccine AMV002 in patients with previously treated HPV-associated OPSCC. In total, 12 patients were assigned to four different groups and received varying dosages of the AMV002 vaccine. The vaccine was tolerated at all dosage levels, and 83.3% of patients demonstrated vaccine-induced cell-mediated response as measured by IFN-γ ELISPOT and ELISA. One subject was noted to have a greater than four-fold increase in antibody response to HPV16 E6 and E7 post-vaccine administration [100].

A phase Ib/IIa trial (NCT02163057) studied the MED10457 DNA vaccine administration in 21 patients with locally advanced HPV16+ HNC. MED10457 expresses HPV 16/18 E6/E7 proteins as well as IL-12. Two separate cohorts were studied, one with administration prior to surgery and another with administration two months post-chemoradiation therapy. Results showed that 18 out of 21 patients generated a durable HPV16 antigen-specific immune response post-MED10457 administration [101]. A follow-up phase Ib/IIa trial (NCT03162224) further investigated the utility of MED10457 administration with concurrent anti-PD-L1 durvalumab therapy. In total, 35 patients with recurrent or metastatic HNC were enrolled, all of whom had received at least one prior platinum-containing regimen or alternative therapy if platinum ineligible. Interim analysis revealed an ORR of 22.2% (6/27 patients). Three patients had a complete response, and an additional three patients had a partial response to therapy. No patients exhibited grade 4/5 treatment-related adverse events, with fatigue and injection site pain noted to be the most common adverse events. Of note, the induction of an immune response was demonstrated by analysis of peripheral HPV-specific T cells and tumor-infiltrating CD8 T cells [102].

The benefits of combining ICI and therapeutic vaccines are an enticing prospect. Another phase Ib/II trial (NCT03260023) investigated TG4001, a modified Ankara vector-based vaccine, in addition to avelumab therapy in patients with HPV-16-positive cancers. Nine patients were included, of which five were patients with HPV-related oropharyngeal cancer. Interim results revealed no serious adverse events, and 33% had a confirmed partial response. Three patients exhibited an increased peripheral T cell response against E6/E7, as assessed by ELISPOT. Subsequent analyses are planned to compare TG4001 with avelumab versus avelumab therapy alone [103].

An additional therapeutic vaccine that has been more extensively studied is the peptide vaccine ISA-101. ISA-101 incorporates long peptide chains derived from E6 and E7. Previous studies in HPV-related cervical cancer have shown that monotherapy with ISA-101 caused an increased T cell response but did not lead to regression or prevention of tumor progression [104,105]. A subsequent phase II trial (NCT02426892) investigated ISA-101 therapy in combination with nivolumab. Twenty-two out of twenty-four patients had HPV-related oropharyngeal cancer. ORR was 33% [106]. A follow-up phase II trial on the combination of ISA-101 and nivolumab is currently ongoing [107]. There are multiple other ongoing studies involving ISA-101, including one phase II study investigating its use in conjunction with utomilumab (a CD137 agonist monoclonal antibody) (NCT03258008) [108] and another phase II study that is alternatively looking at its therapeutic potential in combination with pembrolizumab (NCT04369937) [109].

A phase I/II trial (NCT0523851) is investigating the PDS0101 liposomal peptide vaccine as monotherapy and combined with pembrolizumab in patients with HPV-related OPSCC [110]. Additionally, a phase I/II study (NCT04180215) is analyzing monotherapy of HB-201, a live attenuated arenavirus vaccine that encodes both E6 and E7 oncoproteins and is being utilized in human subjects for the first time. Of note, this study includes patients with recurrent or metastatic HPV-related HNC who have not received treatment and are eligible for pembrolizumab [111].

While several therapeutic vaccines show encouraging results, some trials have shown less promising results. A phase I study (NCT01493154) investigated the use of the pNGVL4a-CRT/E7 DNA vaccine in conjunction with cyclophosphamide in HPV-16-associated HNC patients and was terminated early due to serious adverse reactions [112]. Another phase I, non-randomized study (NCT02002182) is investigating ADXS11-001, a live attenuated Listeria monocytogenes vaccine targeting the HPV16 E7 oncoprotein revealed 55.6% grade 3 or 4 serious adverse reactions, including one death [113]. Trials regarding therapeutic vaccination and additional ongoing studies are are listed in Table 3 [99,100,101,102,103,106,108,109,110,111,112,113,114,115].

Overall the therapeutic vaccination efforts in HPV-related OPSCC are promising and multifaceted. Progress in vaccine development and translation to clinical applications may alter the therapeutic landscape of managing advanced or localized HPV-related OPSCC within the next decade.

## 5. Primary Prevention through Pre-Exposure Vaccination

Despite the many advancements described above, primary prevention through prophylactic vaccine administration should be a primary effort toward reducing the burden of this disease. There are currently three clinically developed vaccines for human papillomavirus, including the quadrivalent vaccine (Gardasil), the 9-valent vaccine (Gardasil-9), and the bivalent vaccine (Cervarix). In the United States, Gardasil-9 is primarily available, which targets HPV types 6, 11, 16, 18, 31, 33, 45, 52, and 58 [116,117]. As described earlier, therapeutic vaccines primarily target oncoproteins E6 and E7. Prophylactic vaccines, on the other hand, target the L1 surface protein, which is the major component of the viral capsid and an integral intermediary for HPV host cell infection [118]. The protective effect of these prophylactic vaccines is thought to be almost exclusively mediated by the induction of neutralizing antibodies that will bind to L1 and thus prevent host cell infection [116].

Currently, no established randomized controlled trials or epidemiological studies have evaluated how prophylactic vaccines impact the incidence or progression of OPSCC related to HPV. Despite this, the potential benefits are assumed, given the effectiveness that has been demonstrated for anogenital and other HPV-associated disease states [119,120,121]. The FDA approved Gardasil for the prevention of HPV-related oropharyngeal and other head and neck malignancies in June 2020 [122].

The efficacy of prophylactic HPV vaccination in impeding HPV-related HNC is limited to various studies analyzing its effects on the reduction of oral HPV disease. A randomized control trial conducted in the United States and Brazil, published in 2018, demonstrated that the quadrivalent HPV vaccine was approximately 88% effective in preventing persistent oral HPV infection amongst HIV-infected adults when compared to a placebo group [123]. These results reinforce the findings of a prior cross-sectional study completed in 2013 that originally investigated vaccine efficacy against cervical cancer in Costa Rican females through a randomized control trial. Of the 7466 females randomized to receive either the bivalent HPV vaccine or the Hepatitis A vaccination, a lower prevalence of oral HPV infection in the HPV-vaccinated group compared to the control group was noted [124]. Another cross-sectional study from 2018 replicated similar findings in the United States, where there was a significant reduction in the prevalence of oral HPV infections in vaccinated versus unvaccinated young adults, with an estimated 88.2% reduction in vaccination status. These results, even though encouraging, have limited effect on the overall population due to low vaccine uptake in the United States [125]. A more recent longitudinal cohort study published in 2019 followed sexually active adolescent females for approximately a decade; they detected significantly fewer HPV-targeted vaccine types in oral samples of individuals who had received at least one dose of the quadrivalent vaccine as compared to unvaccinated subjects [126].

Other retrospective studies have demonstrated that introducing the HPV vaccine into the general populations of the United States and the United Kingdom has decreased the prevalence of HPV-16 infections in unvaccinated individuals, suggesting that there is a herd community component and advantage of prophylactic vaccination for reducing disease burden [127,128]. Amongst these various study designs, the general consensus points toward an apparent reduction in the prevalence of oral HPV infections with vaccination. Whether that also corresponds to subsequent amelioration in HPV-related HNC disease burden requires long-term follow-up and further investigation in future studies.

## 6. Conclusions

Researchers have worked for decades to find new therapies to treat cancer, and immunotherapy now represents a beacon of hope for many types of malignancy. Recent advancements have further supported that metaphor, with PD-1/PD-L1-based immunotherapies positively impacting outcomes in HNC specifically. Studies so far have indicated this benefit to be limited to a small subset of patients, making further research into novel immunotherapeutics even more vital. Such benefits are much needed in HNC as it remains a challenging disease with an overall poor prognosis and limited treatment options.

There are multiple modalities in which immunotherapy is being investigated based on therapeutic targets and emerging biomarkers. HPV-related HNC represents both a unique potential and challenge in that its distinct immune microenvironment may affect how the disease responds to immunotherapy; these differences have and will direct how therapeutic approaches are devised and administered. Future research into this area is exciting as previous studies have demonstrated that HPV-associated disease has increased expression of PD-L1 and immune infiltration.

Understanding these differences further will be essential for developing personalized immunotherapies for HPV-related HNC. Considering how experimental and pioneering the current advancements are, evidence is still limited in terms of how this may impact HPV-related HNC. However, the ongoing development of HPV-specific immune modulators is revolutionary, including therapeutic vaccines and T cell-associated therapeutics targeting HPV. Whether these immunotherapeutics will be sufficient in their own merit or whether combinatorial approaches with ICIs and other anti-cancer agents will be needed are areas that require future exploration.

## Figures and Tables

**Figure 1 cancers-15-01959-f001:**
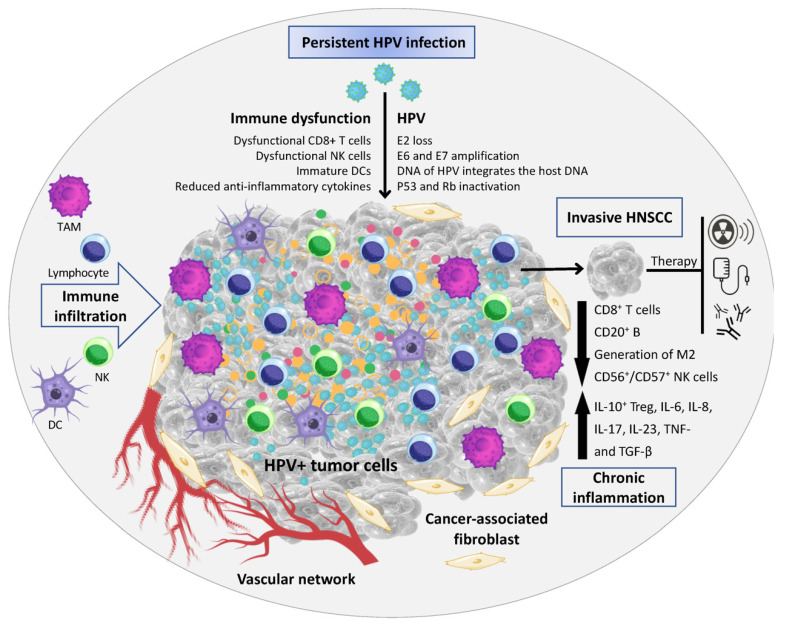
HPV-related head and neck squamous cell carcinoma tumor microenvironment [53]. Key: Figure adopted with permission from SD da Silva, corresponding author of the manuscript “Implications and Emerging Therapeutic Avenues of Inflammatory Response in HPV+ Head and Neck Squamous Cell Carcinoma”. HPV: Human Papillomavirus, HNSCC: Head and neck squamous cell carcinoma, TAM: tumor-associated macrophages, NK: natural killer cells, DC: dendritic cells, IL: interleukin, Treg: regulatory T cell, TNF: tumor necrosis factor, TGF: transforming growth factor.

**Table 1 cancers-15-01959-t001:** Major immunotherapy trials and their outcomes in relation to HPV Status.

Trial	Study Population	Regimen	Patient	Primary Endpoint	Results per HPV Status
KEYNOTE-048 (phase III) [51]	RM, untreated SCC	A: Pembrolizumab B: Pembrolizumab, platinum, 5-FU C: Cetuximab, platinum, 5-FU	A: 301 B: 281 C: 300	OS (B vs. C) CPS > 20: 14.7 vs. 11.0 month CPS > 1: 13.6 vs. 10.4 month	No difference in OS between HPV+ and HPV− patients
CONDOR (phase II) [63]	RM, platinum-refractory with low PD-L1	A: Durvalumab + Tremelimumab B: Durvalumab C: Tremelimumab	A: 133 B: 67 C: 67	A vs. B vs. C ORR: 7.8%, 9.2%, 1.6% Grade ¾ AE: 15.8% vs. 12.3% vs. 16.9%	HPV+ vs. HPV− in three arms ORR: 5.4% vs. 16.7% vs. 0%
KEYNOTE-012 (phase IB) [65]	RM, untreated and treated SCC	Single Arm Pembrolizumab	132	ORR: 18% Any Grade AE: 62% Grade > 3: 9%	HPV+ vs. HPV− ORR: 32% vs. 14% 6-month PFS: 37% vs. 20% 6mo OS: 70% vs. 56%
KEYNOTE-055 (phase II) [66]	RM, platinum-refractory SCC	Single Arm Pembrolizumab	171	ORR: 16% Any Grade AE: 64% Grade > 3: 15%	HPV+ vs. HPV- ORR: 16% vs. 15% 6-month PFS: 25% vs. 21% 6 month OS: 72% vs. 55%
KEYNOTE-040 (phase III) [67]	RM, platinum-refractory SCC	A: Pembrolizumab B: Standard of Care	A: 247 B: 248	OS A: 8.4 month B: 6.9 month	No difference in OS between HPV+ and HPV− patients
CHECKMATE-141 (phase III) [69]	RM, platinum-refractory SCC	A: Nivolumab B: Standard of Care	A: 240 B: 121	OS (1/2 years) A: 36.0%/16.9% B: 16.6%/6.0%	OS benefit noted in all groups irrespective of PD-L1 expression or HPV status
CHECKMATE-651 (phase III) [70]	RM, untreated SCC	A: Nivolumab + Ipilimumab B: EXTREME	A: 472 B: 475	OS (total/CPS > 20/CPS ≥ 1) A: 13.9/17.6/15.7 month B: 13.5/14.6/13.2 month	No difference in OS between HPV+ and HPV− patients
HAWK (phase II) [71]	RM, immunotherapy naïve with high PD-L1	Single arm Durvalumab	111	ORR: 16.2%	HPV+ vs. HPV− ORR: 30% vs. 11.8% Median PFS: 3.6 vs. 1.8 month Median OS: 10.2 vs. 5.0 month
EAGLE (phase III) [72]	RM, platinum-refractory SCCC	A: Durvalumab B: Durvalumab + Tremelimumab C: Standard of Care	A: 240 B: 247 C: 249	OS (A vs. C): 7.6 vs. 8.3 OS (B vs. C): 6.5 vs. 8.3	NA

Key: Standard of Care: Cetuximab OR Methotrexate OR Docetaxel, 5-FU: 5-Fluorouracil, RM: Recurrent/Metastatic, SCC: squamous cell carcinoma, PFS: progression-free survival, ORR: overall response rate, OS: overall survival, HPV: Human Papillomavirus, EXTREME: Cetuximab, platinum-based chemotherapy, and 5-FU.

**Table 2 cancers-15-01959-t002:** Novel/ongoing trials concerning therapeutics targeting E6/E7 oncoproteins.

Trial	Target	Therapeutic Arms	Primary Endpoints/Results Related to HPV-Positive Opscc
NCT01585428 (phase II) [80]	HPV E6/E7 oncoproteins	Young TIL + Fludarabine + Cyclophosphamide + Aldesleukin	OTRR: 18% in (2/11 patients) with noncervical HPV-associated cancer. One patient with oropharyngeal cancer attained a PR of 5 months duration.
NCT02858310 (phase I/II) [82]	HPV E7 oncoprotein	E7 TCR cells + Fludarabine + Cyclophosphamide + Aldesleukin	OTRR: 50% in 12 patients with HPV-related cancers. Four patients with HPV-related oropharyngeal cancer, three had a PR, and one had SD.
NCT03578406 (phase I) [83]	HPV E6 oncoprotein	E6 TCR cells +/− anti-PD1 auto-secreted element	MTD
NCT02280811 (phase I/II) [84]		E6 TCR cells + Fludarabine + Cyclophosphamide + Aldesleukin	MTD, OTRR, DR. 1/12 patients with oropharyngeal cancer. Two patients with PR and four patients with SD, which included the patient with HPV-positive oropharyngeal cancer.
NCT03978689 (phase I) [90]	HPV E7 oncoprotein	CUE-101 +/− Pembrolizumab	DLT, Serum PK parameters. In total, 14 evaluable patients in the monotherapy group, 1 with PR and 6 with SD. In total, 7 evaluable patients in the combination group, 2 with PR and 2 with SD.

Key: Young Tumor Infiltrating Lymphocytes (TIL), partial response (PR), stable disease (SD), T cell receptor (TCR), maximum tolerated dose (MTD), OTRR (objective tumor response rate [complete or partial response]), duration of response (DR), dose-limiting toxicity (DLT).

**Table 3 cancers-15-01959-t003:** Novel/Ongoing trials concerning therapeutic vaccination.

	Intervention/ Treatment	Primary Endpoints/Results Related to HPV-Positive Opscc
NCT00257738 (phase I) [99]	Cohort 1: Peptide vaccine MAGE-A3 Cohort 2: Peptide vaccine GL-0817	Patients with HPV16-positive HNC, no dose-limiting toxicity was observed. 67% (4/6 patients) in the MAGE-A3 arm and 80% (4/5 patients) in the GL-0817 arm developed a T cell and antibody response.
ACTRN12618000140257 (phase I) [100]	DNA vaccine AMV002	Patients with previously treated HPV-positive OPSCC, 83% (10/12 patients) developed a vaccine-induced cell-mediated response. One patient developed a greater than four-fold increase in response post-vaccine administration.
NCT02163057 (phase I/II) [101]	DNA vaccine MED10457	Patients with locally advanced HPV16-positive HNC, 85% (18/21) developed a HPV-16 antigen-specific response
NCT03162224 (phase I/II) [102]	DNA vaccine MED10457 + durvalumab	Patients with recurrent or metastatic HPV-positive HNC, OTRR of 22.2% (6/27 patients), 3 patients had a CR, and 3 patients had a PR.
NCT03260023 (phase I/II) [103]	Modified Ankara vector-based vaccine TG4001 +/− avelumab	5/9 patients with HPV-positive OPSCC, 33% (3/9 patients) with PR.
NCT02426892 (phase II) [106]	Peptide vaccine ISA-101 + nivolumab	22/24 patients with HPV-positive OPSCC, OTRR 33% (8/24 patients). Cure rate in patients with HPV-positive OPSCC 9% (2/22).
NCT03258008 (phase II) [108]	Peptide vaccine ISA-101 + utomilumab	OTRR, Patients with HPV16-positive OPSCC.
NCT04369937 (phase II) [109]	Peptide vaccine ISA-101 + pembrolizumab	PFS, Patients with HPV16-positive HNC.
NCT0523851 (phase I/II) [110]	Liposomal peptide vaccine PDS0101 +/− pembrolizumab	Proportion of successful response, OTRR, PFS, patients with locally advanced HPV-positive OPSCC.
NCT04180215 (phase I/II) [111]	Live attenuated arenavirus vaccine HB-201 +/− pembrolizumab	MTD, OTRR, patients with recurrent/metastatic HPV+ HNC.
NCT01493154 (phase I) [112]	DNA vaccine pNGVL4a-CRT/E7	Patients with HPV16-positive HNC. Enrolled two patients but was terminated early due to serious adverse events.
NCT02002182 (phase I) [113]	Live attenuated Listeria monocytogenes vaccine ADXS11-001	Patients with HPV-positive OPSCC, preliminary results showed 55.6% (5/9 patients) had grade 3 or 4 serious adverse events
NCT02865135 (phase I/II) [114]	Peptide vaccine DPX-E7	Patients with HPV-positive cancers, of which OPSCC is included.
NCT04432597 (phase I/II) [115]	Gorilla adenovirus vaccine PRGN-2009	T cell infiltration response, MTD, Patients with HPV-positive cancers, of which OPSCC is included

Key: OTRR (objective tumor response rate [complete or partial response]), CR (complete response), PR (partial response), PFS (progression-free survival), MTD (maximum tolerated dose).

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
