# Peer review of "Novel Immunotherapeutic Approaches to Treating HPV-Related Head and Neck Cancer"

_cancers, 2023, doi:10.3390/cancers15071959_

Round 1
Reviewer 1 Report
1. Suggest changing the heading type for "B Cell Associated Therapeutics" to italics rather than bold. It currently looks like the vaccination sections are subheadings under B cell therapy, but the transition/relationship is not clear (especially since as stated, vaccines are mostly targeting T cell responses). These may all be able to go under the "Immunotherapy" heading, with a separate bold heading for vaccines (therapeutic and/or primary prevention)
2. The therapeutic vaccine section is quite long and reads too much like a list of trials, with no concluding paragraph. Would benefit from more editorial comments/summary, similar to the final paragraph of the T cell therapeutics section.
3. While not the focus of this HPV+ article, may be helpful to add a short section mentioning other immunomodulatory strategies/targets other than CPI, T cell and vaccination (for example, IDO1, TLR8 agonists, OX40, B7-H3)
Author Response
Reviewer 1
- Suggest changing the heading type for "B Cell Associated Therapeutics" to italics rather than bold. It currently looks like the vaccination sections are subheadings under B cell therapy, but the transition/relationship is not clear (especially since as stated, vaccines are mostly targeting T cell responses). These may all be able to go under the "Immunotherapy" heading, with a separate bold heading for vaccines (therapeutic and/or primary prevention)
We appreciate the reviewer’s comments: The B- cell section was modified to italic as suggested as it is a subheading of the immunotherapy section; as suggested we have kept the vaccine and primary prevention sections separate.
- The therapeutic vaccine section is quite long and reads too much like a list of trials, with no concluding paragraph. Would benefit from more editorial comments/summary, similar to the final paragraph of the T cell therapeutics section.
As suggested this section was trimmed and a conclusive statement was added
- While not the focus of this HPV+ article, may be helpful to add a short section mentioning other immunomodulatory strategies/targets other than CPI, T cell and vaccination (for example, IDO1, TLR8 agonists, OX40, B7-H3)
We have added a brief description at the end of the “immunotherapy” section covering the prospects of using these therapeutic approaches with potential applications in HPV related OPSCC
Reviewer 2 Report
This review by Saba and colleagues is well written and completely describes the use of different immunotherapies in HPV-related OPSCC. Paragraphs are well distributed and give clear notions. I would only suggest few changes to improve the reading. Authors should be more careful in the use of acronyms, since some full name words are missing. For example: line 232 ORR, line 370 HLA, line 38- NK and Treg cells, line 388 TME.
I would also suggest to put the reference of each trial together with the name of the trial in all the tables. Finally, in the 'Immunotherapy' paragraph I would add more recent studies for the use of less toxic strategies in HPV-positive HNC, such as the De-ESCALaTE HPV: " Radiotherapy plus cisplatin or cetuximab in low-risk human papillomavirus-positive oropharyngeal cancer (De-ESCALaTE HPV): an open-label randomised controlled phase 3 trial".
Author Response
Reviewer 2
This review by Saba and colleagues is well written and completely describes the use of different immunotherapies in HPV-related OPSCC. Paragraphs are well distributed and give clear notions. I would only suggest few changes to improve the reading. Authors should be more careful in the use of acronyms, since some full name words are missing. For example: line 232 ORR, line 370 HLA, line 38- NK and Treg cells, line 388 TME.
We appreciate the reviewer’s comments and have corrected the acronyms in the manuscript as suggested
I would also suggest to put the reference of each trial together with the name of the trial in all the tables. Finally, in the 'Immunotherapy' paragraph I would add more recent studies for the use of less toxic strategies in HPV-positive HNC, such as the De-ESCALaTE HPV: " Radiotherapy plus cisplatin or cetuximab in low-risk human papillomavirus-positive oropharyngeal cancer (De-ESCALaTE HPV): an open-label randomised controlled phase 3 trial
We thank the reviewer for these suggestions; the Deescalate even though focused on HPV related OPSCC is not clearly linked to the topic of immunomodulatory approaches in this disease. As the reference of this study should imply the mention of multiple other non-immune focused trials we respectfully disagree with this suggestion.
Reviewer 3 Report
Saba and co-workers wrote a very interesting review on immunoterapy approches to HPV+ NHC. Paper is well organized and clair. However, some minor modification are needed before publication.
1-in the introduction should be clearly reported the aims of review.
2-A figure that describe HPV infection effect on Immune microenvirioment in NHC should be insert to make more clair the paragraph
Author Response
Reviewer 3
Saba and co-workers wrote a very interesting review on immunotherapy approaches to HPV+ NHC. Paper is well organized and clear. However, some minor modification are needed before publication.
1-in the introduction should be clearly reported the aims of review.
We have added a section describing the overall aims of this review.
2-A figure that describe HPV infection effect on Immune microenvironment in NHC should be insert to make clearer the paragraph;
We have added a figure (1) describing the effect on the immune-microenvironment on HPV related OPSCC adapted with permission from the referenced manuscript’s corresponding author.